# A Hybrid Intrusion Detection Model Combining SAE with Kernel Approximation in Internet of Things

**DOI:** 10.3390/s20195710

**Published:** 2020-10-08

**Authors:** Yukun Wu, Wei William Lee, Xuan Gong, Hui Wang

**Affiliations:** 1College of Computer Science and Technology, Zhejiang University of Technology, Hangzhou 310023, China; 1121712004@zjut.edu.cn (Y.W.); 1111712011@zjut.edu.cn (X.G.); 1111712012@zjut.edu.cn (H.W.); 2School of Artificial Intelligence, Zhejiang Post and Telecommunication College, Shaoxing 312366, China

**Keywords:** stacked auto-encoder, support vector machine, random Fourier feature, kernel approximation, intrusion detection, NSL-KDD

## Abstract

Owing to the constraints of time and space complexity, network intrusion detection systems (NIDSs) based on support vector machines (SVMs) face the “curse of dimensionality” in a large-scale, high-dimensional feature space. This study proposes a joint training model that combines a stacked autoencoder (SAE) with an SVM and the kernel approximation technique. The training model uses the SAE to perform feature dimension reduction, uses random Fourier features to perform kernel approximation, and then random Fourier mapping is explicitly applied to the sub-sample to generate the random feature space, making it possible to apply a linear SVM to uniformly approximate to the Gaussian kernel SVM. Finally, the SAE performs joint training with the efficient linear SVM. We studied the effects of an SAE structure and a random Fourier feature on classification performance, and compared that performance with that of other training models, including some without kernel approximation. At the same time, we compare the accuracy of the proposed model with that of other models, which include basic machine learning models and the state-of-the-art models in other literatures. The experimental results demonstrate that the proposed model outperforms the previously proposed methods in terms of classification performance and also reduces the training time. Our model is feasible and works efficiently on large-scale datasets.

## 1. Introduction

Cybersecurity issues in the Internet of Things (IoT) are an increasingly prominent challenge due to the continuous development and sophistication of network penetration methods and attack technologies [1]. How to quickly identify intrusions in the IoT has become one of the urgent problems in the field of network security. It is critical to develop more effective detection technology. Signature-based network intrusion detection systems (NIDSs) and anomaly-based NIDSs are two types of NIDS. A key technical issue is that NIDSs need to identify never-before-seen network attacks. In a signature-based NIDS, the system detects attacks based on pre-installed rules. The NIDS detects the intrusion in the network traffic dataset by comparing the network traffic with the updated attack signature database.

Anomaly-based NIDSs distinguish whether the network traffic is normal or abnormal by learning the abnormal data in the network data stream. Network behavior deviating from the normal data pattern is considered to be abnormal behavior. We have mainly studied anomaly-based NIDSs because of their ability to detect unknown or abnormal attacks. Anomaly detection methods are used in national security departments, banking, finance, securities, communications, and medical applications [2]. In fact, the IDS mainly detects whether the network traffic is normal or abnormal, so that the IDS can be attributed to a classification problem.

Many machine learning algorithms have been applied to IDSs to improve their performance, resulting in breakthroughs [2]. However, traditional machine learning methods work better when dealing with small datasets. They are less efficient when dealing with the classification of massive intrusion data in an actual network application environment. Conventional machine learning algorithms usually have problems with high time complexity and space complexity when large-scale data have high-dimensional properties or nonlinear feature spaces. Therefore, proposing or adopting a more effective method to reduce the dimensionality of high-dimensional data is an indispensable step in the construction of NIDSs.

Deep learning (DL) theory was first proposed by Professor Hinton [3] in 2006. Today, the DL technique is widely used in image recognition, speech recognition, machine translation, and emotion recognition. The advantage of deep learning is that it can obtain meaningful abstract representations from raw data through multilayer neural networks. The significant difference between deep learning and traditional machine learning methods is that it can extract features from raw data instead of using manually designed features. Effective features can greatly improve the intrusion detection performance of NIDS. The feature learning of deep learning is automatically completed through a multi-layer neural network. Each layer of neurons will extract features from the input data and output to the next layer. Compared to traditional machine learning, deep learning reduces the dependence on the prior distribution of the data. As the depth of the network increases, the number of nonlinear transformations of the data increases, and the stronger the model representation ability for data, thereby obtaining a more essential characterization of the data. DL also offers advantages in handling high-dimensional large-scale data. However, a major limitation of SAE is that the loss function is non-convex. Therefore, the model tends to converge to local minimum values and cannot guarantee the existence of the global minimum value. This study proposes an intrusion detection model that combines stacked autoencoder (SAE) with support vector machine (SVM), gaining the advantages of both. There are two main reasons why SVM is theoretically attractive [4,5]: (1) SVMs have good generalization performance when parameters are properly configured, even if there are some deviations in the training set. (2) Since the loss function is a convex function, it provides a unique solution. In principle, SVMs can model any training set when selecting the appropriate kernel. 

Given the complementary advantages of the SVM and SAE models, it makes sense to study a hybrid model of these two architectures for intrusion detection. However, the previous work in this area also focused on hybrid models, but in these, the two components were trained separately, unlike in a true hybrid model. First, the DL model is trained to obtain the compression features after a dimensionality reduction, and then the extracted features are fed to the shallow model to continue training. For example, the hybrid model of a deep belief network (DBN) and One-class SVM (OC-SVM) proposed in [6] uses a DBN to compress high-dimensional features, and uses the traditional OC-SVM for anomaly detection. However, the two-stage approach does not effectively learn the characteristics of the intrusion detection dataset, and the dimensionality grows larger due to the decoupling of the learning phase. Therefore, the kernel approximation technique is applied to approximate the kernel function of nonlinear SVM. The feature extracted by the SAE is used to construct the explicit feature map approximation for the Gaussian kernel. The linear SVM algorithm is then used to efficiently solve the large-scale nonlinear SVM. The SAE is combined with the linear SVM to implement end-to-end training.

The main contributions of this study are as follows:(1)Through the kernel approximation technique, SAE and SVM are combined for joint training, and a new intrusion detection model is proposed. The effectiveness and the potential of our method for handling large-scale data are studied to improve upon traditional supervised machine learning algorithms.(2)We studied the impact of the network structure of SAE and the number of random Fourier features on the accuracy of our method.(3)We used the NSL-KDD dataset for experimental verification, and compared the efficiency of our approach with non-kernel approximation SVM and different classification algorithms in binary-category and five-category classification. The experimental results confirm that our model is more accurate and faster.

The remainder of this study is given as follows: Section 2 outlines other work related to this study. Section 3 offers background for the related technologies used in this study and discusses the proposed intrusion detection model in greater detail. Section 4 describes the setup of the experiment. Section 5 provides the experimental results and comparative analysis. In Section 6, we present our conclusions and discuss potential future research.

## 2. Related Work

Researchers have previously used various traditional machine learning methods to construct IDS, such as SVM [7], the k-nearest neighbor (KNN) algorithm [8], the naive Bayes (NB) algorithm [9], Artificial Neural Networks (ANN) [10], Random Forest (RF) [11], the self-organizing map (SOM) type of ANN [12], and other related machine learning methods [13,14]. In all these methods, the performance of SVM was improved [15]. For example, the results in [16] demonstrate that SVM was better than ANN in terms of test time and accuracy on the KDD CUP 99 dataset. Kou et al. [17] compared the performance of traditional machine learning methods such as SVM, logistic regression (LR), naive Bayes, decision tree (DT) and classification and regression tree (CART) on the KDD CUP99 dataset in terms of accuracy and false alarm rate. The experiment results confirmed that SVM had a distinct performance. Although traditional machine learning algorithms and shallow learning technologies (e.g., ANN) have improved detection accuracy and reduced false alarm rates, the detection performance of these methods is highly dependent on features. With Google’s artificial intelligence (AI) program AlphaGo defeating South Korean Li Shishi in chess, DL has become a research hotspot in the field of AI. As a branch of machine learning, DL can learn suitable and effective features from a vast amount of complex data. The DL model effectively solves the “curse of dimensionality” by learning low-dimensional feature representations and achieving good detection results in high-dimensional and complex network environments. There are already many DL methods applied in IDSs, such as the deep convolutional neural network (DCNN) [18,19], recurrent neural network (RNN) [20], long short-term memory network (LSTM) [21], and DBN [22]. However, since the loss function of DL is non-convex, it will also face the limitation of converging to a local minimum. Therefore, many researchers use a combination of DL and other different classification algorithms such as SVM and RF to construct an IDS.

Alom et al. [23] used a DBN and a SVM to construct a new intrusion detection model. The role of the DBN is to remove redundant features and noisy data and extract valid features, and then the SVM is used for classification. Although this model achieved better results than a single DBN and a single SVM, the DBN and SVM are trained separately rather than jointly trained together. Cao et al. [24] used an autoencoder (AE) to extract the features of the intrusion dataset and then used the density estimation method for classification. Similarly, pertaining to large-scale training data, Shone et al. [25] used a stacked nonsymmetric deep autoencoder (SNDAE) for unsupervised feature learning, and combined RF algorithms for intrusion detection. The results show that the model had higher accuracy, precision, and recall, and reduced training time. Compared with the method based on DBN, the model accuracy was increased by 5%, and the training time was reduced by up to 98.81%. AL-QATF et al. [26] used a single-layer sparse AE to remove redundant dimensions and extract effective features, and replace softmax with SVM for classification. Experimental results showed that the model improved the SVM classification performance and reduced time spent. Good performance was also shown to have been achieved in the binary and five categories on the NSL-KDD dataset, compared to the other models. Zhang et al. [27] proposed a real-time DBN-SVM intrusion detection model, and achieved good detection results on the CICIDS2017 dataset. However, the training methods of DBN and SVM in this model are separately trained.

Qiao et al. [28] established an Internet of Things (IoT) network detection model based on a deep autoencoder (DAE) and an SVM. The DAE is used to learn the feature representation of the data, and then that is classified by a powerful SVM. The SVM uses the artificial bee colony (ABC) algorithm and the fivefold cross-validation (5FCV) statistical method to find the optimal parameters. The experimental results on the UNSW-NB15 dataset show that the model had better performance than some advanced intrusion detection methods, including principal component analysis (PCA) and other different classification algorithms. In [29], an intrusion detection model combining a gated recurrent unit (GRU) and linear SVM was proposed. The structure of GRU RNN was improved, and the SVM was used as a classifier. In addition, the SVM replaced the edge-based function with the cross-entropy function. The results on the Kyoto2013 dataset show that the performance of the GRU_SVM model achieved significant improvement over the traditional GRU model. Chowdhury et al. [30] trained a DCNN as a feature extractor to extract the output of each layer of the CNN as the input of the SVM. This method was inspired by small sample learning and solves the sample imbalance problem. Experimental verification was carried out on the KDD99 and NSL-KDD datasets, and the results show the effectiveness of the proposed model.

The above hybrid model by Chowdhury et al is basically a combination of a deep neural network (DNN) and an SVM. The two can complement each other, but the above models are trained separately. This training method cannot effectively learn the feature representation of sample data. Moreover, when the SVM with Gaussian kernel function is used as a classifier, the required computation is very complex, and the big data problem cannot be solved effectively. Therefore, this paper proposes an intrusion detection model combining a stacked AE and a kernel approximate linear SVM. The model adopts the joint training mode and approximates the Gaussian kernel with random Fourier features, which not only solves the problem of large-scale data processing, but also reduces the training time.

## 3. Materials and Methods

### 3.1. Stacked Autoencoder (SAE)

The standard AE is a multilayer feedforward network with structural symmetry on the middle layer, including input layer, hidden layer, and reconstruction layer, as shown in Figure 1. The entire AE consists of an encoder and a decoder.

The first part of the AE is the encoder: the process of mapping input to an implicit representation. The second part of the AE is the decoder: the process of mapping an implicit representation to an output layer to reconstruct the input.

Given an *N*-dimensional training set $D=(x_{1},x_{2},\cdots ,x_{n-1},x_{n})$, where $x_{n}$ is the n-th vector in the training set. After the training set D is trained by the AE, the output vector ${\bar{x}}_{n}$ of the output layer will reconstruct the input vector $x_{n}$ as much as possible.

The process of these two parts is as shown in Equations (1) and (2):(1)$$y_{n}=f({Wx}_{n}+b)$$
(2)$${\bar{x}}_{n}=g(W^{*}y_{n}+c)$$
where W and b represent weights and bias in the encoding stage, $W^{*}$ and c represent weights and bias in the decoding stage, and f and g are activation functions in the encoding and decoding stages, respectively.

The SAE superimposes multiple AEs using the representation of the hidden layer of the previous layer as the input of the next layer, resulting in a more abstract representation, as shown in Figure 2. The SAE initializes the network weight parameters and bias by greedy layer-wise pre-training [31], thereby increasing the convergence speed of the deep network and slowing down the effects of gradient disappearance. The SAE is then fine-tuned using the error backpropagation method until the reconstruction error is minimized, thereby obtaining an optimal parametric model.

The SAE is an unsupervised neural network model whose core function is to learn the deep representation of input data. The input layer data x is converted to its hidden layer representation and then reconstructed by the hidden layer to restore new input data. The training goal of the SAE is to make the reconstructed data restore the input layer data as much as possible. The loss function of the SAE is usually defined by mean square error, as shown in Equation (3):(3)$$L(W,b,c)=\frac{1}{2}\sum_{k=1}^{n}{\left(x_{k}-{\bar{x}}_{k}\right)}^{2}$$

To avoid overfitting, the weight decay is usually added to the loss function, and (3) is rewritten as (4):(4)$$L(W,b,c)=\frac{1}{2}\sum_{k=1}^{n}{\left(x_{k}-{\bar{x}}_{k}\right)}^{2}+L_{wd}$$

The equation for weight decay is expressed as follows:(5)$$L_{wd}=\frac{1}{2}\lambda \left(|\right|W{\left|\right|}_{F}^{2}+\left|\right|W^{*}{\left|\right|}_{F}^{2})$$
where $\left|\right|\cdot {\left|\right|}_{F}$ is the *F*-norm of the matrix and λ is the weight decay coefficient.

Using an optimization algorithm such as gradient descent, the optimal parameters (W,b,c) of the model can be determined.

### 3.2. SVM

A SVM is based on maximizing an interval and finding a hyperplane that correctly classifies the data. Since the kernel strategy is used, SVMs can usually capture nonlinearity. The kernel strategy plays a significant role in solving nonlinear inseparable mode analysis. The main idea of the kernel strategy is to map a low-dimensional linearly inseparable feature space $R^{d}$ to a high-dimensional space $R^{D}$. It is expected that the data are linearly separable in the high-dimensional space. The commonly used kernel functions mainly include linear kernel functions, radial basis kernel functions, and polynomial kernel functions. This paper uses a radial basis kernel function (RBF) $K(x_{i},x_{j})=exp(-\left|\right|x_{i}-x_{j}{\left|\right|}^{2}/2\sigma^{2})$. Assume that the training data is ${\left\{\left(x_{i},y_{i}\right)\right\}}_{i=1}^{l}$, $x_{i}\in R$, $y_{i}\in \left\{-1,+1\right\}$, then the object of the nonlinear SVM is to form a problem for solving convex quadratic programming under certain constraint conditions. The formulas are as shown in (6) and (7):(6)$$min\frac{1}{2}{\left||\hat{W}|\right|}^{2}+C\sum_{i=1}^{n}\xi_{i}$$
(7)$$s.t.\;y_{i}({\hat{W}}^{T}\phi (x_{i})+b)\geq 1-\xi_{i}$$
where $\hat{W}$, b, C, $\xi_{i}$, and $\phi (x_{i})$ represent the weight vector, bias term, penalty factor, slack variable, and kernel function, respectively. In general, one can replace ${\hat{W}}^{T}\phi (x_{i})+b$ with ${\hat{W}}^{T}\phi (x_{i})$ by adding one dimension to include bias item b in $\hat{W}$. In addition, using hinge loss instead of $\xi_{i}$, the unconstrained objective function of the nonlinear SVM is as shown in (8):(8)$$min\frac{1}{2}\left|\right|\hat{W}{\left|\right|}^{2}+C\sum_{i=1}^{n}max(0,1-y_{i}{\hat{W}}^{T}\phi (x_{i}))$$

Equation (8) is essentially a convex optimization problem that can be solved by Lagrangian duality. However, dual space is directly proportional to the number of samples in the dataset. Assuming that the number of samples is n, the time complexity will be $O(n^{2})$.This is intolerable for large datasets.

Although there are few efficient algorithms for solving nonlinear SVM, there are many efficient algorithms for solving linear SVMs. In recent years, the feature has been explicitly used as the input of linear SVMs, which can transform the solution of a nonlinear SVM into a linear SVM in the feature space. Instead of solving the convex quadratic optimization problem, the kernel-induced feature space is embedded into a relatively low-dimensional explicit stochastic feature space, and the linear SVM is trained so that the efficient linear SVM algorithm can be applied to solve the nonlinear SVM.

### 3.3. Kernel Approximation Random Fourier Feature

The kernel approximation algorithm is often used to solve the scalability problem of kernel machine learning. There are two primary kernel approximation algorithms: Nystroem and random Fourier feature (RFF) [32,33]. This paper uses RFF since it has low complexity and does not require pre-training. Bochner’s theorem [34] states that as long as the kernel function satisfies shift-invariance and is continuous and positive definite, there is a Fourier transform with a probability distribution p(⋅) corresponding to the k(⋅) function. As shown in Equation (9):(9)$$k(x,y)=\int_{R^{d}}p(w)e^{jw^{T}(x-y)}dw$$where p(w) is the probability density function of w, for the Gaussian kernel function, p(w) can be calculated by the inverse Fourier transform of k(x,y) and we obtain w~N(0,2γI), where I represents the identity matrix.

D independent and identically distributed weights $w_{1},w_{2},\ldots ,w_{D}$ are drawn from the distribution p(w) by Monte Carlo sampling. For sample x, its random Fourier features z(x) are defined as follows (10):(10)$$z(x)=\sqrt{\frac{1}{D}}{[\cos(w_{1}^{T}x)\ldots \cos(w_{D}^{T}x)\sin(w_{1}^{T}x)\ldots \sin(w_{D}^{T}x)]}^{T}$$where $w_{i}\in R^{d}$ is subject to normal distribution N(0,2γI). Applying kernel approximation mapping, the unconstrained objective function of the nonlinear SVM is transformed into the following:(11)$$\frac{1}{2}||\hat{W}|{|}^{2}+C\sum_{i=1}^{n}max(0,1-y_{i}{\hat{W}}^{T}z(x_{i}))$$

In this way, the training set is converted to $\{(z(x_{i}),y_{i}){\}}_{i=1}^{l}$ and as the input of the linear SVM, the efficient solving algorithm of the linear SVM can be applied to the nonlinear SVM. The optimal solution of the objective function will become simpler, although the dimension of $R^{D}$ is higher than the dimension of $R^{d}$.

### 3.4. Proposed Method For Intrusion Detection

#### 3.4.1. The Proposed Model Framework

The overall framework of the proposed model based on the SAE and the kernel approximation SVM is shown in Figure 3, which includes five steps: Step 1:Data preprocessing. The training set and test set are respectively processed for numeralization of symbolic data and normalization; see Section 4.1.2 for details.Step 2:The training set X is first fed to the SAE, and forward calculation is performed to obtain a compressed representation. Then, the compressed representations with kernel approximation (KA) are used as the input of the linear SVM.Step 3:Calculate the reconstructed data $\bar{X}$ and the loss function.Step 4:Use the gradient descent method for joint training to obtain a training model. See the next section for details.Step 5:Make a prediction on the test set.

#### 3.4.2. Joint Training Model

The joint training model based on SAE and kernel approximation SVM is shown in Figure 4. The model is made up of an SAE, which is used for dimensionality reduction and compressed representation of the original data. The SVM with radial basis kernel function uses the random Fourier feature for kernel approximation.

In the following, we describe the model in detail.

Assume that the input vector of the SAE is X and the reconstructed vector is $\bar{X}$. x is the feature representation of the hidden layer after dimension reduction. In addition, θ is a set of parameters for the SAE and the weight matrix $\hat{W}$ is the main parameter of the SVM. Accordingly, the objective function of the model is shown in Equation (12):(12)$$Q(\theta ,\hat{W},X)=\alpha L_{dae}+L_{svm}$$

The hyperparameter α is the adjustment factor between the feature dimension reduction of the SAE and the interval optimization of SVM, $L_{dae}$ as shown in Equation (4), $L_{svm}$ as shown in Equation (13):(13)$$L_{svm}=\frac{1}{2}||\hat{W}|{|}^{2}+C\sum_{i=1}^{n}max(0,1-y_{i}{\hat{W}}^{T}z(x_{i}))$$

$z(x_{i})$ is the Fourier feature map of the SAE compression feature, as shown in Equation (9) above.
(14)$$g(x)={\hat{W}}^{T}z(x)=\sum_{j=1}^{D}w_{j}z_{\omega j}(x)\;=\sqrt{\frac{1}{D}}\sum_{j=1}^{D}[w_{j}cos(w_{j}^{T}x)+w_{D+j}sin(w_{j}^{T}x)]\;=\sqrt{\frac{1}{D}}\sum_{j=1}^{D}\left[w_{j}cos(\sum_{k=1}^{d}w_{jk}x_{k}\right)+w_{D+j}sin(\sum_{k=1}^{d}w_{jk}x_{k})]$$

g(x) represents the interval function of the SVM. Each $x_{k}$ in the equation represents the feature representation of each neuron in the hidden space of the last layer of the encoder, and $x_{k}=f(\theta ,X)$, f is the activation function of the SAE. The gradient of g(x) with respect to $x_{k}$, θ can be derived as:(15)$$\frac{\partial g}{\partial x_{k}}=\sqrt{\frac{1}{D}}\sum_{j=1}^{D}w_{jk}[-w_{j}\sin(\sum_{k=1}^{d}w_{jk}x_{k})+w_{D+j}\cos(\sum_{k=1}^{d}w_{jk}x_{k})]$$
(16)$$\frac{\partial g}{\partial \theta }=\frac{\partial g}{\partial x_{k}}\frac{\partial x_{k}}{\partial \theta }$$

The gradient of the model loss function with respect to the SAE parameters and the SVM weight vector is as shown in Equations (17) and (18):(17)$$\frac{\partial Q}{\partial \theta }=\frac{\partial L_{dae}}{\partial \theta }+\frac{\partial L_{svm}}{\partial \theta }$$
(18)$$\frac{\partial Q}{\partial W}=\frac{\partial L_{svm}}{\partial W}$$

Finally, combining (17) and (18) with (14), we can obtain an end-to-end gradient of loss function with respect to all model parameters based on the chain rules of the derivative.

To implement this model, we use the TensorFlow open-source machine learning platform to perform automatic differentiation, to obtain the necessary gradients to minimize the loss function in the training process, and to process large datasets.

The joint training process of the model is described by Algorithm 1.

In Algorithm 1, *N* represents the number of samples and *r* represents the number of SAE hidden layers. $u_{j}^{n}$ and $h_{j}^{n}$ respectively represent the input and output of the *n*-th sample in the *r*-th hidden layer.
**Algorithm 1**: Joint training algorithm.Input: Training set $D=\left\{\left(x_{n},y_{n}\right)|1\leq n\leq N\right\}$, SAE structure, number of training iterations t(0≤t≤T)Output: Model parameters $\theta =\left\{W^{j},b^{j},1\leq j\leq r\right\}\cup^{\;}\left\{\hat{W}\right\}$Training stage 1. Randomly initialize $W^{j}$ and $b^{j}$, $\hat{W}$. 2. *t* = 0. 3. While (*t* ≤ *T*)  1) Forward calculation to obtain the compressed representation of SAE:   $h_{0}^{n}=x_{n},u_{j}^{n}=W^{j}h_{j-1}^{n}+b^{j},h_{j}^{n}=f\left(u_{j}^{n}\right)(1\leq j\leq r)$  2) Perform kernel approximation according to Equation (10), get $z\left(h_{r}^{n}\right)$.  3) Calculate *g*(*x*) according to Equation (14), where $z\left(h_{r}^{n}\right)$ is the input of the SVM.  4) Use the gradient descent method to obtain model parameters $\left(\theta ,\hat{W}\right)$ according to Equations (15)–(18).  *t* = *t* + 1  End 4. Get the optimal parameters of the model: θ={W,b} and $\hat{W}$. 5. According to Equations (10) and (14), calculate *g*(*x*) on the test set.  6. Determine whether the sample is normal or malicious based on the value of *g*(*x*).

## 4. Experiment Setup

### 4.1. Dataset and Preprocessing

#### 4.1.1. Dataset

The NSL-KDD [35] dataset is formed on the basis of KDD99, and solves some problems in KDD99. The training set in the NSL-KDD dataset has no redundant data, so the classifier will not be biased to more frequent records; there is no repeated data in the test set, so the final detection rate will be more accurate; moreover, in the NSL-KDD dataset, the sizes of the training set and test set are more reasonable, such that it can be used directly during the experiment without having to randomly select a small part of it. The NSL-KDD dataset contains four sub-datasets: NSL-KDDTrain+, NSL-KDDTrain_20percent, NSL-KDDTest+, and NSL-KDDTest21. Each connection record in the dataset has 41 features, the first to the tenth features contain the basic information of the network connection, the 11th to the 22nd features contain the content information of the network connection, and the 23rd to the 41st features contain flow feature information. The test set contains 17 attack types that do not appear in the training set, and the purpose is to detect the generalization ability of the model. Attack types are divided into the following four categories: Dos, Probe, R2L, and U2R. The details are shown in Table 1. It has recently been used in a large number of security research projects [36,37,38].

#### 4.1.2. Data Preprocessing

1. Numeralization

Each sample has 41-dimensional features and 1 label. The second, third, and fourth-dimensional features are all symbolic data, and the remaining features are numerical features. 

Among them, the second-dimensional feature is the protocol type, the third-dimensional feature is the network service, and the fourth-dimensional feature is the flag, and one-hot encoding is performed for these symbolic features. We can use one-hot encoding to represent the protocol types TCP, UDP, and ICMP as (1, 0, 0), (0, 1, 0), (0, 0, 1), respectively. Finally, the 41-dimensional features are transformed to 122-dimensional features.

2. Normalization

Some of the features of the NSL-KDD dataset have a very large range, such that feature values cannot be compared and are not suitable for processing. Therefore, all the feature values are mapped to the [0, 1] range by max-min normalization.

### 4.2. Evaluation Metrics

The confusion matrix is a universally applicable tool in machine learning which can help researchers to understand the errors in the classification and evaluate the quality of the classifier. The confusion matrix used in this paper is shown in Table 2. In the paper, *TP* is the number of attack samples that are correctly identified as attack; *FP* is the number of normal samples that are wrongly identified as attack; *TN* is the number of normal samples that are correctly identified as normal; *FN* is the number of attack samples that are wrongly identified as normal. The total number of test samples is *S* = *TP* + *FN* + *FP* + *TN*, where the correctly predicted number of samples is *TP* + *TN*, and the number of incorrectly predicted samples is *FP* + *FN*. The general sample classification problem evaluation indicators mainly include Accuracy (*AC*), Precision (*P*), Recall (*R*), Detection rate (*DR*), F-score (*F1*), and False alarm rate (*FAR*). *DR* and *FAR* are as follows:(19)$$DR=\frac{TP}{TP+FN}$$
(20)$$FAR=\frac{FP}{FP+TN}$$

## 5. Discussion

To verify the validity of the proposed method, we designed two sets of experiments. The purpose of the experiments was to make a comparison in the performance of the joint training model with kernel approximation (JSAE-FSVM), the separate training model with kernel approximation (SSAE-FSVM), and the separate training model without kernel approximation (SSAE-SVM) in binary-category classifications (Normal, Abnormal) and five-category classifications (Normal, DoS, R2L, U2R, and Probe). At the same time, to compare our proposed model with several conventional machine learning methods (NB, RF, KNN, DT, SVM, and MLP(multi-layered perceptron)), we designed a set of comparative experiments. The experiments in the study were designed to realize four objectives:(1)Evaluate the effect of AE structure on model performance.(2)Evaluate the effect of Fourier features number on model performance.(3)Evaluate the binary-category and five-category performance of the joint training model.(4)Analyze the advantages of the proposed model.

The experimental environment for this study was as follows: We used the extensive DL framework TensorFlow for programming. The experiment was conducted on a laptop equipped with Intel Core i7-7700HQ 2.80 GHz CPU, 8 GB RAM, and the 64-bit Windows 10 operating system, without GPU acceleration. The parameters of the SAE were set as follows: the activation function was a tanh function; batch size was 256; weight initialization was per Xavier’s method [39]; learning rate was 0.001. The adjustment factor in Equation (12) was 10,000. 

### 5.1. Impact of the Different Network Structures

The determination of the structure of the DNN primarily involves the determination of the depth of the networks and the width of each hidden layer. Currently, there is no mature theoretical method to select the optimal network structure for as DNN. If the model is too simple, it may not be able to effectively extract the compressed representation of the input vector—that is, it differs greatly from the probability distribution of the original data, and it cannot denote the essential representation of the original data. Conversely, deeper models mean better non-linear representation capabilities, which can learn more complex transformations, which can fit more complex feature inputs. However, if the model is too complex, the training process becomes more complex, which will dramatically increase the space-time consumption and produce overfitting. Accordingly, different network structures need to be set according to different experimental environments.

Table 3 shows the experimental results in the binary-category classifications when using different network structures in the NSL-KDD test set.

The experimental results show that as the number of hidden layers increased, the classification effect improved, but the amount of time spend on training also increased, which is not feasible in the massive intrusion data environment. In the above experiment, the training time of the five-layer SAE was 554.79 s, while the training time of the other two four-layer structures was 510.06 s and 474.52 s, respectively. When the model faces a large-scale intrusion data environment, the time complexity will increase exponentially. According to the test results of the SAE on the NSL-KDD training set in Table 3, the SAE network structure selected in this paper was [122-110-85-55-15].

### 5.2. Impact of Fourier Feature Numbers

This paper evaluates the effect of random Fourier feature numbers (sample feature numbers) on the proposed model’s accuracy. We performed experiments with different feature numbers on the NSL-KDDTest+ and NSL-KDDTest21 datasets. Figure 5 shows the classification accuracy and training time for feature numbers from 100 to 450.

As can be seen from Figure 5, as the number of features increased, the performance of our model improved. The more sampling dimensions, the better the classification results, but the greater the cost. This means that there is a trade-off between run-time and accuracy, and the number of features selected in this paper was 400.

### 5.3. Classification Performance of the Model

To verify the superiority of our proposed network intrusion detection model (JSAE-FSVM), we measured the performance of our model in the binary-category and five-category classifications in NSL-KDDTest+ and NSL-KDDTest21. We compared this performance with that of the separately trained classification model (SSAE-FSVM) and the separate training model (SAE-SVM) without kernel approximation in terms of performance index and training time. The results are shown in Table 4 and Table 5.

It can be seen from Table 4 and Table 5 that all JSAE-FSVM performance metrics, except for precision and *FAR*, were higher than those for the other two models. In the binary-category experiment, especially on the NSL-KDDTest21 dataset, the accuracy of our model was as high as 74.4%, which is about eight percentage points higher than the other two models, a remarkable increase. In the five-category model, the accuracy rate on the NSL-KDDTest+ dataset was better than on the other two. The *FAR* of JSAE-FSVM was lower than that of SSAE-FSVM and SAE-FSVM for both binary-category and five-category experiments. The most attractive aspect of the model is that the training time of the model with approximate Gaussian kernel showed a sharp drop compared to the others. This is mainly because we used efficient linear support vector machines, which are very important in the NIDS in large-scale datasets.

### 5.4. Comparison with Other Models

#### 5.4.1. Comparison with Basic Models

We also compared the accuracy of other classification algorithms in related research to verify the superiority of our model. Many algorithms have been applied to intrusion detection, including traditional machine learning algorithms such as NB, KNN, DT, RF, SVM, and MLP, and DL algorithms such as DBN, SAE, CNN, and RNN. We compared the performance of the model in terms of accuracy with other classification algorithms discussed in [40] and [41], as shown in Figure 6 and Figure 7.

As can be seen from Figure 6 and Figure 7, the performance of the DL algorithms, including RNN and DBN, was superior to that of all conventional machine learning algorithms. In the binary-category experiment, the accuracy of our model was 2.5% and 5.8% higher than that of the methods in [40] on NSL-KDDTest+ and NSL-KDDTest21. Compared with the DBN and MLP methods, the accuracy of the model in the NSL-KDDTest+ dataset increased by 3.5% and 4.6%, respectively, and increased by 6.9% and 16.9% in the NSLKDDTest21 dataset. In the five-category experiment, the accuracy of our model was 2.2% and 3.9% higher than that of the methods in [40] on NSL-KDDTest+ and NSL-KDDTest21. Compared with the DBN and MLP methods, the model proposed in this paper increased by 3% and 4.6%, respectively, on the NSL-KDDTest+ dataset and by 4.2% and 6.8%, respectively, on the NSL-KDDTest21 dataset.

As can be seen from the above experiments, the accuracy of the proposed method was very close to or exceeded those of other state-of-the-art methods, particularly in the binary-category experiment. This is mainly because we use SAE technology to reduce the dimension, then perform kernel approximation on the features after the dimension reduction, and then finally use the linear SVM for joint training.

*DR* and *FAR* are two important evaluation indicators in IDS. Table 6 shows the overall performance of traditional machine learning methods and basic deep learning methods on the NSL-KDDTest+ and NSL-KDDTest21 datasets. From the table, it can be seen that JSAE-FSVM was superior to the basic classifiers in terms of *DR* and *FAR*.

#### 5.4.2. Comparison with State-of-the-Art Models

In order to further verify the superiority of the proposed model, we compared the proposed model with related models in other literature, including SCDNN [42], LSTM_4_ [43], GRU_3_ [43], CFBLS [43], TSE-IDS [44], ROS-DNN [45], SMOTE-DNN [45], and ADASYN-DNN [45]. To be fair, all the methods were implemented on NSL-KDDTest+ and NSL-KDDTest21. The relevant index result of some methods was not given, and we have used N/A to indicate this. Detection indicators included *DR*, *AC*, and *FAR*. The comparison results are shown in Table 7.

Table 7 shows that, in terms of the overall accuracy of the model, the results were superior to other models, and only slightly lower than the TSE-IDS model in [43].This fully shows that our model fully mined the essential features of the data through end-to-end training, and conducted effective learning and training on it, so as to improve the model’s overall accuracy. In terms of the FAR, our model also achieved good results, with 0.051 being the lowest among all models.

Based on the comparison, it can be seen that the model proposed in this paper was not only superior to the traditional machine learning models and deep learning models, but also had some advantages compared with the latest models in other works. The main reasons are as follows: (1) The adoption of joint training can make it possible to learn the essential features of data and improve the accuracy of the model. (2) The kernel approximation technique and linear SVM were used to accelerate the training speed of the model.

## 6. Conclusions

Traditional machine learning methods have low detection performance and cannot meet the requirement of real-time characteristic when an IDS faces a large-scale intrusion detection dataset. This study proposes a joint training model that combines an SAE with an SVM and kernel approximation. We verified the validity of the model on the NSL-KDD dataset. First, this study used the SAE model to perform unsupervised dimension reduction on samples. As a result, the dimension of the samples dropped from 121 to 15. Then, the 15-dimensional compressed representation was subjected to random Fourier kernel approximation, and then sent to the linear SVM for end-to-end training. In the whole processing process, the SAE reduced the scale of data, and random Fourier features and linear SVM accelerated the processing process.

The application of sparse constraints to the hidden layer enables the model to learn the best feature representation of the samples in high-dimensional and large-scale data environments, and can effectively reduce the dimension of the sample, which improves the generalization ability of the model and its feature extraction capabilities.

The experimental results show that the proposed model is highly suited for large-scale, high-dimensional task scenarios, reduces the training time and test time for the intrusion detection model, can meet the real-time requirement of intrusion detection, and provides superior detection performance to traditional machine learning algorithms. The real-time performance of the model complies with the real-time rules of vector 2 in the IoT cyber risk vectors proposed in [46].

The proposed intrusion detection model is feasible and efficient, and provides a new research direction for intrusion detection. 

In future work, we will carry out our research from two aspects: (1)We will deploy the model in the cloud computing environment in accordance with the network security assessment framework standard in [46], while adopting the latest artificial intelligence technology to design an intrusion detection system for the Internet of Things to make it more aligned with the network security standards in [46].(2)We plan on continuing to study the performance of the model with other kernel approximation methods, as well as the performance of the joint training model with one-class SVMs in anomaly detection.

## Figures and Tables

**Figure 1 sensors-20-05710-f001:**
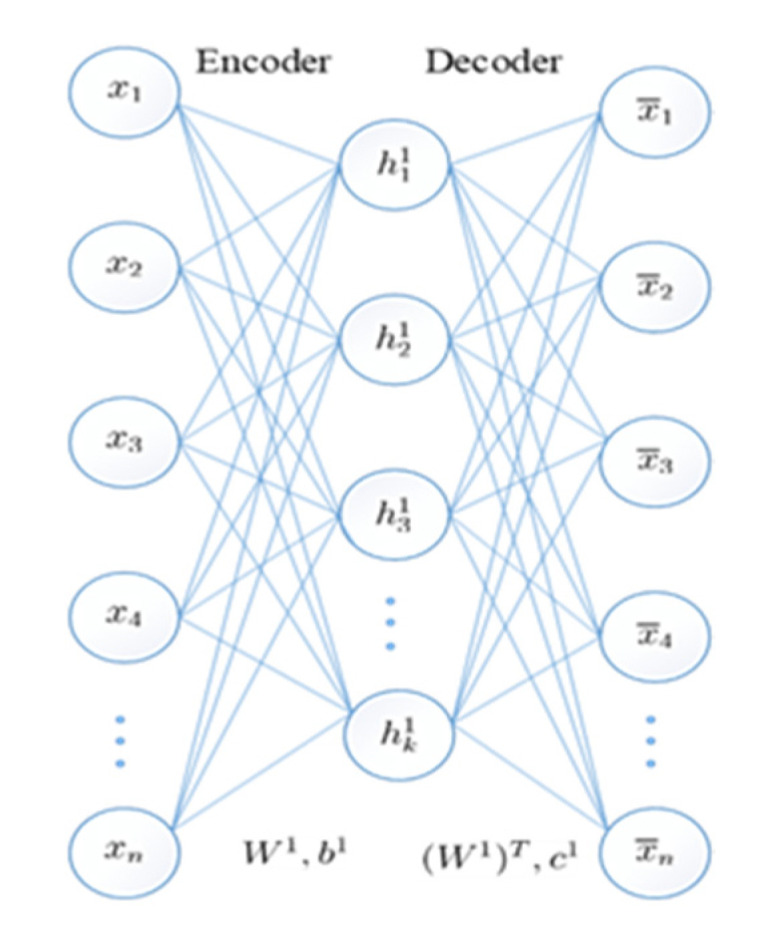
Basic structure of the autoencoder (AE).

**Figure 2 sensors-20-05710-f002:**
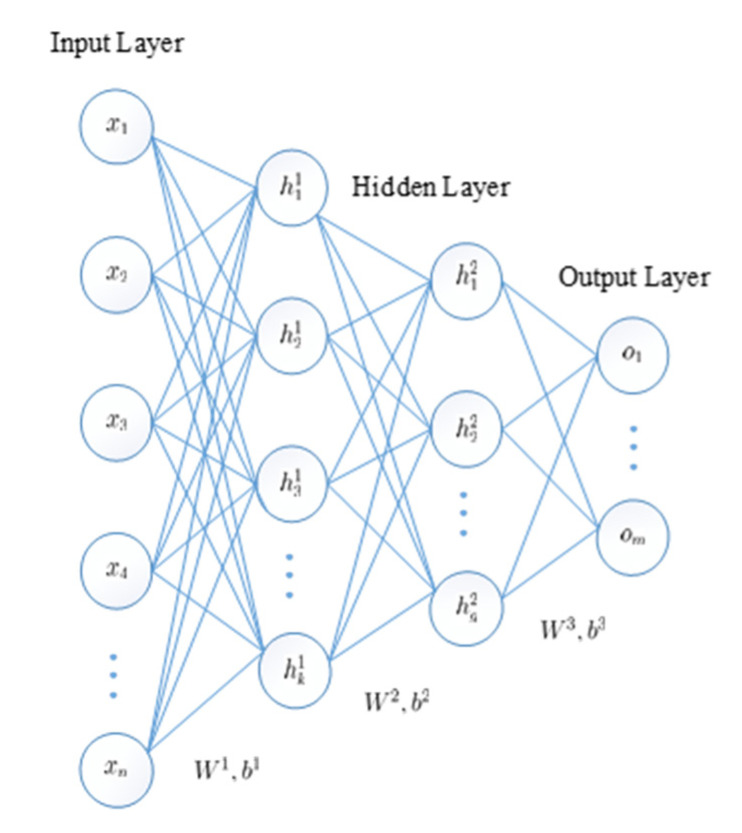
Stacked autoencoder (SAE) structure.

**Figure 3 sensors-20-05710-f003:**
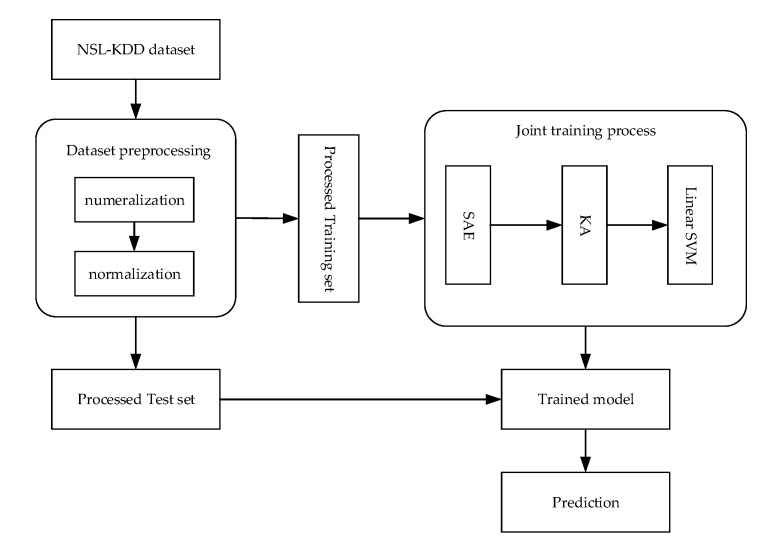
Proposed model framework. (KA denotes kernel approximation)

**Figure 4 sensors-20-05710-f004:**
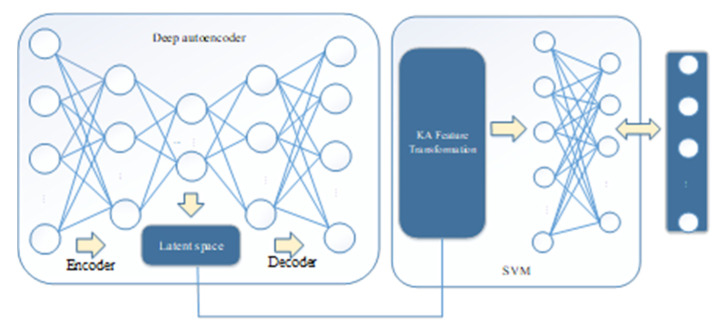
Proposed joint training model.

**Figure 5 sensors-20-05710-f005:**
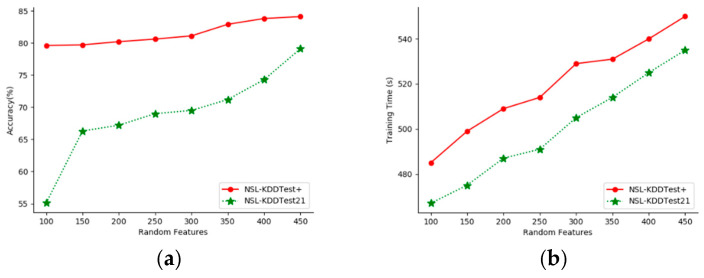
Experimental results with random features from 100 to 450. (**a**) Accuracy. (**b**)Training time.

**Figure 6 sensors-20-05710-f006:**
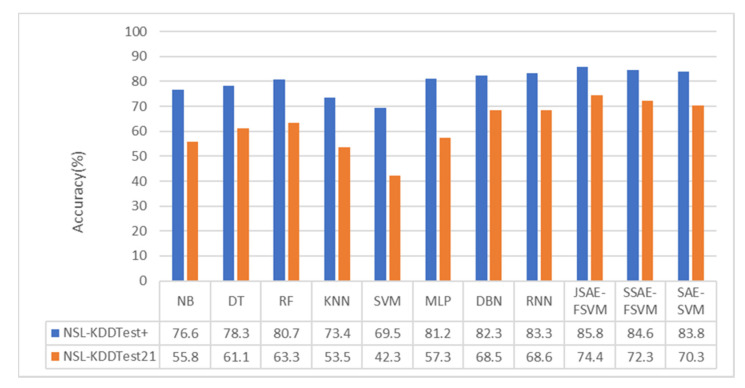
Binary-category classification accuracy.

**Figure 7 sensors-20-05710-f007:**
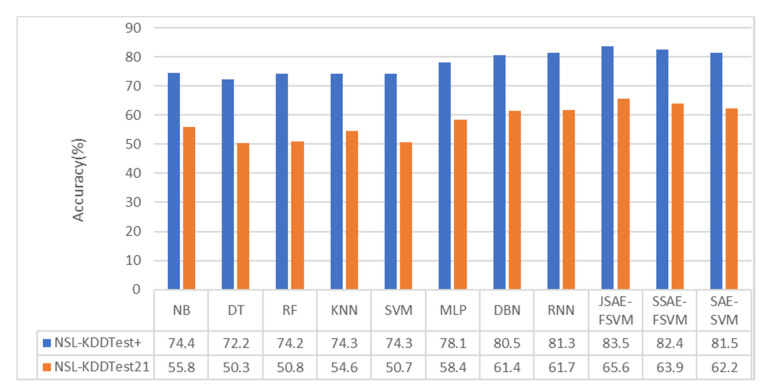
Five-category classification accuracy.

**Table 1 sensors-20-05710-t001:** Distribution of different data types.

Type		NSL-KDDTrain+	NSL-KDDTest+	NSL-KDDTest21
**Normal**		67,343	9711	2152
**Attack**	DoS	45,927	7458	4342
	Probe	11,656	2421	2402
	R2L	995	2754	2754
	U2R	52	200	200
**Total**		125,973	22,544	11,850

**Table 2 sensors-20-05710-t002:** Confusion matrix.

True Category	Predicted Category	
	Attack	Normal
**Attack**	*TP*	*FN*
**Normal**	*FP*	*TN*

**Table 3 sensors-20-05710-t003:** Experimental performance of different network structures.

Network Structure	Accuracy	Recall	Precision	*F1*	*FAR*	Training Time (s)
[122,110,85,55,30,15]	0.842	0.849	0.869	0.859	0.043	554.79
[122,110,85,55,15]	0.838	0.785	0.919	0.857	0.046	510.06
[122,110,80,40,20]	0.837	0.812	0.891	0.850	0.051	474.52
[122,110,58,13]	0.823	0.792	0.884	0.836	0.059	397.53

**Table 4 sensors-20-05710-t004:** Performance in binary classification tests for three hybrid models.

Model	Accuracy	Precision	Recall	*F1*	*FAR*	Training Time (s)
	Binary-category classification—NSL-KDDTest+					
**JSAE-FSVM**	0.858	0.919	0.785	0.847	0.046	510.23
**SSAE-FSVM**	0.846	0.953	0.721	0.821	0.049	532.57
**SAE-SVM**	0.838	0.896	0.756	0.820	0.053	2213.98
	Binary-category classification—NSL-KDDTest21					
**JSAE-FSVM**	0.744	0.815	0.884	0.882	0.186	487.06
**SSAE-FSVM**	0.663	0.945	0.547	0.693	0.193	525.45
**SAE-SVM**	0.593	0.943	0.535	0.683	0.198	756.68

**Table 5 sensors-20-05710-t005:** Performance in five-category classification tests for three hybrid models.

Model	Accuracy	Precision	Recall	*F1*	*FAR*	Training Time (s)
	Five-category classification—NSL-KDDTest+					
**JSAE-FSVM**	0.835	0.823	0.801	0.784	0.051	516.75
**SSAE-FSVM**	0.824	0.774	0.729	0.751	0.063	561.36
**SAE-SVM**	0.815	0.760	0.707	0.733	0.069	1046.20
	Five-category classification—NSL-KDDTest21					
**SSAE-FSVM**	0.656	0.681	0.635	0.647	0.214	492.83
**SSAE-FSVM**	0.639	0.653	0.626	0.639	0.243	545.87
**SAE-SVM**	0.622	0.639	0.601	0.619	0.356	1051.08

**Table 6 sensors-20-05710-t006:** Comparison of the proposed model and the basic models in terms of detection rate (*DR*) and false alarm rate (*FAR*).

Model	NSL-KDDTest+		NSL-KDDTest21	
	*DR*	*FAR*	*DR*	*FAR*
**LR**	0.626	0.072	0.506	0.317
**KNN**	0.667	0.073	0.558	0.316
**DT**	0.644	0.071	0.529	0.320
**RF**	0.605	0.067	0.478	0.302
**SVM**	0.567	0.072	0.427	0.317
**MLP**	0.708	0.074	0.584	0.295
**DBN**	0.658	0.068	0.547	0.304
**CNN**	0.687	0.070	0.607	0.279
**RNN**	0.697	0.069	0.647	0.269
**JSAE-FSVM**	0.801	0.051	0.635	0.214
**SSAE-FSVM**	0.729	0.063	0.626	0.243
**SAE-SVM**	0.707	0.069	0.601	0.356

**Table 7 sensors-20-05710-t007:** Comparison with the state-of-the-art models.

Model	NSL-KDDTest+			NSL-KDDTest21		
	*AC*	*DR*	*FAR*	*AC*	*DR*	*FAR*
**SCDNN** [42]	0.726	0.575	N/A	0.446	0.379	N/A
**LSTM_4_** [43]	0.828	N/A	N/A	0.667	N/A	N/A
**GRU_3_** [43]	0.829	N/A	N/A	0.654	N/A	N/A
**CFBLS** [43]	0.822	N/A	N/A	0.675	N/A	N/A
**TSE-IDS** [44]	0.858	0.868	0.117	0.725	N/A	0.180
**ROS-DNN** [45]	0.783	0.740	0.077	0.634	0.657	0.347
**SMOTE-DNN** [45]	0.812	0.676	0.077	0.653	0.572	0.337
**ADASYN-DNN** [45]	0.801	0.698	0.069	0.578	0.601	0.306
**JSAE-FSVM**	0.835	0.801	0.051	0.656	0.635	0.214
**SSAE-FSVM**	0.824	0.729	0.063	0.639	0.626	0.243
**SAE-SVM**	0.815	0.707	0.069	0.622	0.601	0.356

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
