# Peer review of "A Hybrid Intrusion Detection Model Combining SAE with Kernel Approximation in Internet of Things"

_sensors, 2020, doi:10.3390/s20195710_

Round 1
Reviewer 1 Report
In the paper, the author proposes a joint training model that combines a stacked autoencoder (SAE) with an SVM and the kernel approximation technique,but there are still some problems to be improved.
1. In this paper,the authors suggest that the performance of SVM was improved,but there was no mention of it in Related work.Please introduce it in Related work. 2.There is not enough of the description about the proposed method for intrusion detection.
3. The based on the methods are well analyzed, but can be compared appropriately with other existing methods.
Reviewer 2 Report
Paper describes A Hybrid Intrusion Detection Model Combining SAE With Kernel Approximation in Internet of Things.
Subject is modern, relevant and interesting. Paper is well written.
Some comments:
- One well known weakness of anomaly detection based NIDS is high amount of false alerts (false positives). It would be relevant to compare that capability in your results. You do the comparison of Accuracy, Precision etc, but the False Positive Rate would be interested to compare also.
- The equations 19-22 are so basic knowledge in the field of NIDS development that are those relevant to be presented in the text? Same comment for the equation18, I think that all the readers know what is equation for max-min normalization.
Reviewer 3 Report
Good work.
- The conclusion is a bit short, you should consider adding a few sentences describing your findings.
- Introduction is also a bit long, so maybe you could just restructure, take some of the text from the introduction, and edit for the conclusion. Would be interesting to see a short discussion in one or two sentences on how does your findings on intrusion detection relate to standardisation of risk in IoT systems e.g.:https://doi.org/10.1007/s42452-019-1931-0 and maybe a short discussion in one sentence on the future trends in cyber risk analytics and artificial intelligence in IoT systems.
- but these are just suggestions. Your work deserves publication even without these minor edits, so I will recommend accept with minor changes.
